# Antibody Levels at 3-Years Follow-Up of a Third Dose of Measles-Mumps-Rubella Vaccine in Young Adults

**DOI:** 10.3390/vaccines10010132

**Published:** 2022-01-17

**Authors:** Patricia Kaaijk, Alienke J. Wijmenga-Monsuur, Hinke I. ten Hulscher, Jeroen Kerkhof, Gaby Smits, Mioara Alina Nicolaie, Marianne A. van Houten, Rob S. van Binnendijk

**Affiliations:** 1Centre for Infectious Disease Control, National Institute for Public Health and the Environment (RIVM), 3721 MA Bilthoven, The Netherlands; alienke.wijmenga@rivm.nl (A.J.W.-M.); hinke.ten.hulscher@rivm.nl (H.I.t.H.); jeroen.kerkhof@rivm.nl (J.K.); gaby.smits@rivm.nl (G.S.); alina.nicolaie@rivm.nl (M.A.N.); rob.van.binnendijk@rivm.nl (R.S.v.B.); 2Spaarne Academy, Spaarne Gasthuis, 2133 TM Hoofddorp, The Netherlands; MvanHouten2@spaarnegasthuis.nl

**Keywords:** measles-mumps-rubella (MMR) vaccine, mumps outbreaks, antibody response, humoral immunity, seroprotection, waning immunity

## Abstract

Mumps outbreaks and breakthrough infections of measles and rubella have raised concerns about waning of vaccine-induced immunity after two doses of measles-mumps-rubella (MMR) vaccination. In the present follow-up study, serum IgG antibodies against mumps, measles and rubella, as well as the functional neutralizing antibodies against both the mumps vaccine strain and mumps outbreak strains were measured longitudinally in young adults that received a third MMR (MMR3) dose. The mumps-specific IgG and virus neutralizing antibody levels at 3 years after vaccination were still elevated compared to pre-vaccination antibody levels, although the differences were smaller than at earlier timepoints. Interestingly, subjects with low antibody levels to mumps before vaccination benefited the most as they showed the strongest antibody increase after an MMR3 dose. Three years after an MMR3 dose, all subjects had antibody levels to measles and rubella above the internationally agreed antibody cutoff levels for clinical protection. Our data support the recommendation that an MMR3 dose may provide additional protection for those that have become susceptible to mumps virus infection during outbreaks. MMR3 also resulted in an increase in anti-measles and rubella antibody levels that lasted longer than might have been expected.

## 1. Introduction

Over the last two decades, many mumps outbreaks have been reported among highly vaccinated populations in various countries. Most affected individuals during these outbreaks are young adults who were vaccinated as children and did not acquire immunity to mumps by natural infection. The increased risk of mumps in vaccinated young adults appears to be associated to antibody levels declining to unprotective levels of immunity [1,2,3,4,5,6]. Since waning of vaccine-induced immunity is considered a major cause of the resurgence of mumps among vaccinated young adults, studies have been initiated to evaluate the effect of an additional third measles-mumps-rubella vaccine [MMR3] dose on mumps antibody levels [7,8]. Based on these studies, an extra MMR3 dose is anticipated to be a good and safe intervention for controlling a mumps outbreak. In fact, during an outbreak, students who received an MMR3 dose had a lower risk of mumps compared to students who received two MMR doses [9]. However, this immunity boost is likely to be temporary, and the question of how long an MMR3 dose can provide protection against mumps remains.

Antibody levels to the mumps vaccine component have been reported to decline more rapidly compared to antibody levels to the measles and rubella vaccine components [10]. Although waning of measles and rubella antibody levels also occurs after childhood vaccination, no large measles and rubella outbreaks have been reported among highly vaccinated populations, but only among unvaccinated populations [11,12]. This indicates that vaccine-acquired immunity lasts longer when compared to mumps. Nevertheless, vaccinated health care workers who treat measles cases are at risk and have occasionally been involved in measles outbreaks or clusters [13]. Therefore, vaccinated persons with waning immunity to the measles virus may be at an increased risk of contracting measles in settings with high virus exposure and could benefit from receiving an extra MMR3 dose. Furthermore, several studies indicate that a few percent of fully vaccinated young adults lack protected levels of antibodies against rubella [14,15,16]. Rubella virus infection during pregnancy can result in miscarriage, fetal death, or a constellation of congenital anomalies (congenital rubella syndrome [CRS]). Therefore, nonpregnant postpubertal women with low rubella titers may also benefit from an MMR3 dose to boost immunity to rubella [15,17]. Recently, we have described the dynamics of the antibody response to measles, mumps, and rubella viruses in young adults up to 1 year after receiving an MMR3 dose [8,16]. In the present follow-up study, levels of antibodies to the three viruses were evaluated 3 years post-vaccination. This study shows that an additional MMR dose may protect against measles, mumps, and rubella virus infection for longer than previously assumed.

## 2. Materials and Methods

### 2.1. Study Details

In the Netherlands, two MMR doses at the age of 14 months and 9 years have been included in the National Immunization Program since 1987. According to this immunization program, all included participants (*n* = 147) aged 18–25 years had received the two MMR doses in childhood, and none of them had a history of mumps. A flow scheme for the number of participants and study handling at each visit is provided in Appendix A. During home visits, blood samples (clot tube volume, 8 mL) were collected for mumps virus-specific antibody measurements: just before, 4 weeks ± 7 days, and 1 year ± 1 month after an MMR3 dose [8]. At 3 years ± 3 months after vaccination, participants received an instruction on how to perform a finger prick blood sample and self-collect this in a microtainer. Blood samples were returned in safety envelopes to the RIVM laboratory. Serum was separated from the clot and stored at −20 °C until use.

The research protocol was approved by the Dutch ethics committee (EudraCT 2016-001104-36 and Netherlands Trial Register [NTR] NTR5911) and was performed in accordance with The Code of Ethics of the World Medical Association (Declaration of Helsinki) for experiments involving humans. All participants provided written informed consent prior to any study handling.

### 2.2. Determination of Antibody Responses

Serum IgG antibodies for each of the 3 vaccine components were determined in parallel by a fluorescent bead-based multiplex immunoassay, as previously described [8,16]. Briefly, purified measles virus (strain Edmonston [in-house]), purified mumps vaccine strain (Jeryl Lynn [in-house]), and rubella virus (strain HPV-77; GenWay) were used as antigens. For each assay, a reference (RUBI-1–94, calibrated against the international standards for measles and an in-house standard for mumps), controls, and blanks were included. Antibody concentrations were obtained by interpolation of the mean fluorescent intensity in the reference serum curve by using a five-parameters logistic (5PL) regression model and expressed in international units per mL (IU/mL) for measles and rubella and RIVM units (RU/mL) for mumps. An antibody concentration of ≥0.12 IU/mL for measles [18] and ≥10 IU/mL for rubella [19] was used as the cutoff for seroprotection for clinical protection. No international agreement or an accurate serological correlate of protection exists for mumps. Therefore, we used a surrogate level of protection of ≥102 RU/mL, which was previously assessed as the most appropriate cutoff for seroprotection against mumps virus infection [8].

Mumps virus-neutralizing antibodies were determined by a focus-reduction neutralization test (FRNT) as previously described. The FRNT was performed by using either the Jeryl Lynn vaccine strain (genotype A; hereafter, “the vaccine strain”) or the wild-type outbreak strain, MuVi/Utrecht. NLD/40.10 (genotype G; hereafter, “the outbreak strain”), isolated from a throat swab specimen from an individual with mumps [20]. The mumps virus–neutralizing antibody titer was expressed as the dilution of serum that resulted in 50% plaque reduction (hereafter, “the ND_50_ value” [i.e., 50% neutralization dose]), by using the modified Kärber formula [20,21]. The World Health Organization international standard Rub-1–94 was used as reference serum in each FRNT run. To correct for interassay differences, we normalized the ND_50_ values by multiplying the raw ND_50_ values by the Rub-1–94 factor (defined as the cumulative geometric mean value of Rub-1–94 covering all FRNT runs of the study [ie, an ND_50_ value of 850 against the vaccine strain and 1028 against the outbreak strain]) and dividing the product by the measured ND_50_ value of Rub-1–94 in that particular FRNT run. For a valid FRNT, the measured ND_50_ value of the reference Rub-1–94 standard was required to be within 2 SDs of its cumulative mean value. The lower limit of detection for ND_50_ measurements was set to 4 and ND_50_ values ≤ 4 or lower were treated as left-censored. The censoring amounted to 1.1% in the FRNT vaccine strain data and to 2.9% in the FRNT outbreak strain data. Data from samples with at least two valid test results per virus strain in separate test runs were used, only in case of insufficient amount of finger prick serum also test results from one valid test was accepted. Not for all participants the FRNT could be performed at 3 years, due to insufficient volume of finger prick serum that was collected at that time point.

### 2.3. Statistics

Linear effects models were employed to assess the effects of MMR3 dose and sampling time on longitudinal log-transformed neutralizing antibody titers separately for the two mumps virus strains (Figure 1) and separately for measles, mumps, and rubella log-transformed IgG concentrations (Figure 2). Model selection was performed by means of likelihood ratio test. The final 3 models for measles, mumps, and rubella-specific IgG concentrations, as well as the final 2 models for neutralizing antibody titers against the vaccine and outbreak strains, were specified as main effects and as random effects (random intercept and slope) on the four fixed sampling times. The main effects structure allows us to distinguish differences within the individual courses of antibody levels measured across time. The random effects structure, with a specific intercept and slope per individual, allows us to identify differences between the individual course of antibody levels over time as well as their correlates. Based on the results of the model fit, we express fold changes of antibody levels for any combination of time points. In fitting the model, the censored observations were treated as actual measurements, which enabled us to use a more flexible software package. However, an approach in which we accounted for censored observations resulted in similar quality of estimation as the results of the model fit. Using the delta-method, the variances of transformed fold-changes, other than those between the subsequent time points and baseline, were approximated. The setup and application of the linear mixed model were performed using R, version 3.4.3 [22], and visualization was performed using package ggplot2 [23].

Differences in antibody levels between the various time points were analyzed with Wilcoxon signed-rank test (Figure 3). *p*-Values < 0.05 were considered significant. Statistical analyses were performed using Graphpad Prism 9.1.0 [24].

## 3. Results

In the present follow-up study, levels of antibodies to mumps, measles and rubella viruses were evaluated three years after an MMR3 dose and compared with previously published antibody levels at earlier points in time. From a total of 147 young adults (mean age at baseline [range], 22 [18,19,20,21,22,23,24,25] years; male sex, 46%; female sex, 54%), antibody levels were determined prior to vaccination, and at 4 weeks, 1 year and 3 years post-vaccination. At 3 year post-vaccination, 119 out of 147 (81%) participants returned (mean age, 25 [21,22,23,24,25,26,27,28] years; male sex, 40%; female sex, 60%). On average, participants received their second childhood MMR immunization 13 years prior to study entry.

Eighty-one percent of the participants had mumps-specific serum IgG concentrations above the cutoff level for seroprotection at baseline. Mumps-IgG-based seroprotection rates increased to 94% at 4 weeks post-vaccination, and seroprotection rates declined to 90% at 1 year and 87% at 3 years post-MMR3. Similar dynamics of seroprotection rates were observed based on functional (virus-neutralizing) antibody levels against both the mumps vaccine and mumps outbreak viral strains (Appendix A).

In Figure 1, the dynamics of the mumps-specific antibody levels are visually presented by using a linear mixed model. In general, a clear increase was observed at 4 weeks, followed by a steep decline at 1 year and a more gradual decline to a steady antibody level at 3 years after vaccination. Four weeks after vaccination, the increase in mumps virus-specific IgG antibodies was sharper compared to virus neutralizing (VN) antibody levels (Mumps IgG, 1.6-fold increase [lower limit of 95% CI, 1.5 and upper limit of 95% CI, 1.8]; vaccine-VN 1.4-fold [1.3–1.5]; outbreak-VN, 1.4-fold [1.2–1.5]). However, the decline in IgG antibody levels was also more prominent than the levels of VN antibodies (Mumps IgG, 0.84-fold change [0.78–0.90]; vaccine-VN, 0.89-fold [0.83–0.94]; outbreak-VN, 0.97-fold [0.89–1.1]). Mumps IgG antibody concentrations declined more gradually from 1 year to 3 years post-vaccination (Mumps IgG, 0.87-fold change [0.82–0.93]), while VN antibody levels stabilized from 1 year to 3 years after an MMR3 dose (vaccine-VN, 0.99-fold [0.91–1.1]; outbreak-VN, 0.99-fold [0.90–1.1]). The mumps-specific IgG and VN antibody levels at 3 years after vaccination were marginally, though statistically significantly, higher than antibody levels before MMR3 receipt (Figure 1 and Figure 3, Appendix A).

IgG antibody concentrations to measles and rubella viruses showed a similar dynamic pattern of antibody levels after vaccination: a clear increase at 4 weeks, a steep decline at 1 year and a gradual decline to a more stable level at 3 years after vaccination (Figure 2, Appendix A). However, in contrast to mumps, 100% seroprotection levels were maintained for measles and rubella at 3 years after an MMR3 dose (Appendix A). In line with this, the measles and rubella-specific antibody levels at 3 years were still significantly higher than at baseline (Figure 2 and Figure 3, Appendix A). Four weeks after vaccination, the increase in rubella virus-specific IgG antibodies was sharper compared to measles and mumps virus-specific antibodies (Rubella IgG, 3.0-fold [2.6–3.4], Measles IgG, 1.8-fold [1.6–2.0], Mumps IgG, 1.6-fold [1.5–1.8]). However, rubella-specific IgG antibody levels from 4 weeks to 1 year post-vaccination also showed a stronger decrease compared to measles and mumps-specific antibodies (Rubella IgG, 0.59-fold [0.54–0.64]; Measles IgG, 0.84-fold [0.77–0.91]; Mumps IgG, 0.84-fold [0.78–0.90]). IgG antibody concentrations gradually declined to more stabilized levels from 1 year to 3 years post-vaccination (Rubella IgG, 0.75-fold [0.71–0.18]; Measles IgG, 0.86-fold [0.82–0.89]; Mumps IgG, 0.87-fold [0.82–0.93]).

**Figure 1 vaccines-10-00132-f001:**
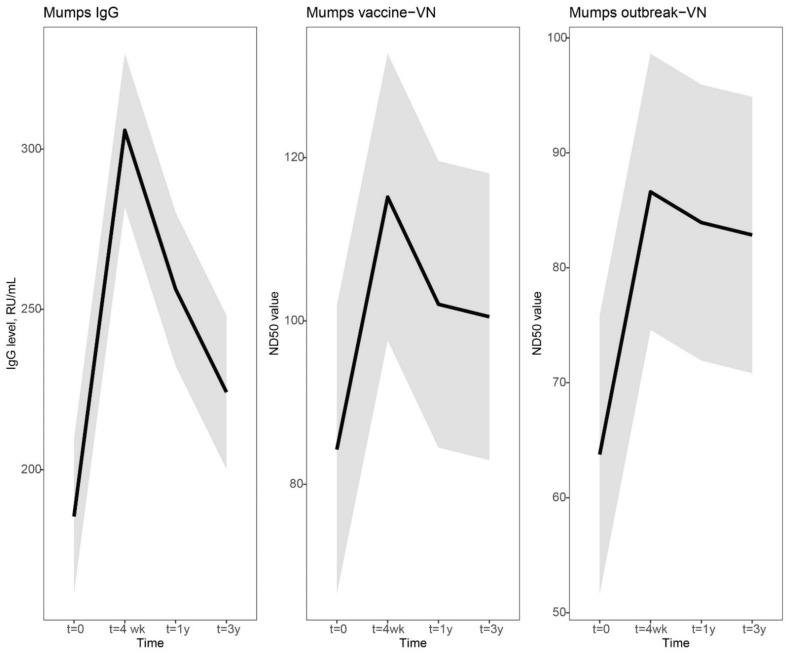
Dynamic course of IgG and virus neutralizing antibody responses to mumps virus after a third measles-mumps-rubella vaccine dose. We used a linear mixed model to compare the dynamics in IgG and virus neutralizing antibody responses to mumps virus after a third measles-mumps-rubella vaccine dose. Fitted averaged dynamic course of antibody levels, based on the results of the model fit, are presented as bold lines. Mumps virus-specific IgG concentrations in RU/mL (left), virus neutralizing antibodies (VN) in ND50 values of focus-reduction neutralization test against mumps vaccine strain (middle), and mumps outbreak strain (right) prior to, and 4 weeks, 1 year and 3 years following an MMR3 dose. Abbreviations: IgG level, mumps virus-specific IgG concentration; RU/mL, RIVM units per milliliter; VN, virus neutralizing antibodies; ND50 values, virus-neutralizing antibody titer which resulted in 50% plaque reduction; t = 4 wk, four weeks after a third dose of measles-mumps-rubella vaccine dose (MMR3); t = 1 y and t = 3 y, resp. 1 year and 3 years after an MMR3 dose.

**Figure 2 vaccines-10-00132-f002:**
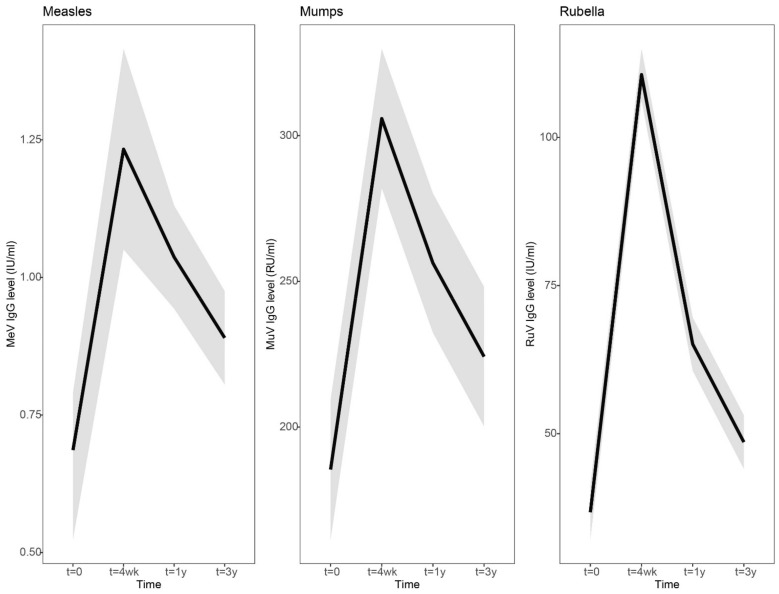
Dynamic course of IgG antibody responses to measles, mumps and rubella virus after a third measles-mumps-rubella vaccine dose. A linear mixed model was used to compare the dynamic IgG antibody response to measles (left), mumps (middle), and rubella (right) prior to, and 4 weeks, 1 year and 3 years following an MMR3 dose. Fitted averaged dynamic course of antibody levels, based on the results of the model fit, are presented as bold lines. Abbreviations: MeV-IgG, measles virus-specific IgG concentration; MuV-IgG, mumps virus-specific IgG concentration; RuV-IgG, rubella virus-specific IgG concentration; RU/mL, RIVM units per milliliter; IU/mL, international units per milliliter; t = 4 wk, four weeks after a third dose of measles-mumps-rubella vaccine dose (MMR3); t = 1 y and t = 3y, resp. 1 year and 3 years after an MMR3 dose.

Interestingly, the lower the pre-vaccination antibody levels against mumps virus were, the stronger were the increases in mumps-specific antibodies from baseline to 1 year and from baseline to 3 years after an MMR3 dose. Based on the results of the model fit, these estimated correlations between low pre-vaccination mumps-specific antibody levels and stronger antibody increases were statistically significant. This implies that persons with antibody levels below the cutoff level and thus at risk for mumps virus infection could clearly benefit from an MMR3 dose.

For the levels of antibodies against measles and rubella viruses we could not gather enough evidence to assess the significance of correlations between pre-vaccination antibody levels and fold-increase at 1 or 3 years post-vaccination due to the lack of variance for the random effects of these data.

**Figure 3 vaccines-10-00132-f003:**
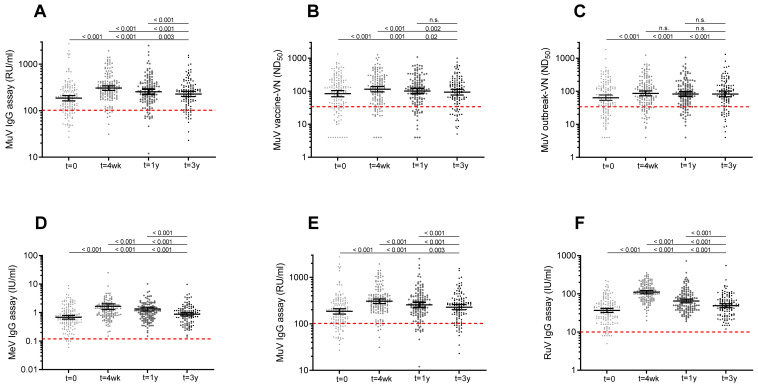
Scatterplot showing IgG and virus neutralizing antibody levels to mumps virus and concentrations of IgG antibodies against rubella and measles virus prior to and after a third measles-mumps-rubella vaccine dose. MuV-IgG concentrations (**A**), ND50 titers against MuV vaccine strain (**B**), and ND50 titers against MuV outbreak strains (**C**) prior to, and 4 weeks, 1 year and 3 years following an MMR3 dose. Concentrations of IgG antibodies against measles (**D**), mumps (**E**), and rubella (**F**) virus prior to, and 4 weeks, 1 year and 3 years after an MMR3 dose. Geometric mean IgG concentrations and ND50 titers, with 95% confidence interval are indicated as horizontal bars. Red dashed line indicate the surrogate antibody cutoff level for protection. Differences in antibody levels between time points were analyzed with Wilcoxon signed-rank test. Observed significant differences were in line with the results of the model fit. P-values related to comparisons to the pre-vaccination antibody levels are listed at the bottom, above are the P-values related to comparisons to the antibody levels at 4 weeks, and at top, the P-value for comparison of antibody levels at 1 year and 3 years after MMR3. Abbreviations: MuV IgG, mumps virus-specific IgG concentration; RU/mL, RIVM units per milliliter; MuV vaccine-VN, virus neutralizing antibodies against Jeryl Lynn (JL) mumps vaccine strain; ND50, virus-neutralizing antibody titer which resulted in 50% plaque reduction; MuV outbreak-VN, virus neutralizing antibodies against mumps outbreak strain; MeV IgG, measles virus-specific IgG concentration; U/mL, units per milliliter; RuV IgG, rubella virus-specific IgG concentration; t = 0, prior to a third dose of measles-mumps-rubella vaccine dose (MMR3); t = 4 wk, four weeks; t = 1 y, one year; t = 3 y, three years after MMR3.

## 4. Discussion

A third measles-mumps-rubella vaccine (MMR3) dose is expected to be a good and safe intervention for controlling a mumps outbreak [9]. Previously, we showed that mumps-specific antibody levels were still higher at 1 year after an MMR3 dose compared to pre-vaccination levels [8]. Three serological estimates were used to evaluate the dynamics of antibody boosting to mumps virus following MMR3, i.e., mumps-specific IgG antibody concentrations, and the functional 50% virus neutralization dose (ND_50_) against both the vaccine and outbreak mumps viruses. Here, we show that 3 years after MMR3 vaccination, mumps-specific antibody levels are still statistically significantly higher compared to pre-vaccination levels, albeit marginally. This result proved unambiguous regardless of which of the 3 serological estimates was used.

An effective immune response induced by vaccination is dependent on the generation and maintenance of B-cell mediated immunological memory [25]. The clear increase in mumps-specific antibodies after an MMR3 dose indicates the presence of a recall response of memory B cells. Interestingly, VN antibody levels seem to have stabilized already between 1 and 3 years after an MMR3 dose. The persistence of functional antibodies suggests that long-lived plasma cells have been formed after an MMR-3 dose.

Since there is no international agreement for a serological correlate of protection for mumps, we here used surrogate cutoff levels of protection for the 3 serological assays. These cutoff levels were previously assessed, and were based on pre-outbreak serum samples comparing antibody levels of persons with and without serological evidence of mumps virus infection during the last mumps outbreak in the Netherlands [8]. According to these surrogate levels, seroprotection rates at 3 years after an MMR3 dose were 87%, 82% and 85%, for respectively mumps-specific IgG antibodies, neutralizing antibodies against the vaccine strain, and outbreak strain, whereas 81%, 78% and 78% of participants were estimated to be protected before vaccination. Previously, Anderson et al. calculated that the critical vaccination coverage needed to block mumps virus transmission was 90–92% [26]. On the assumption of a maximum vaccine effectiveness of 96% against mumps for 2 doses of MMR vaccine [27], the herd immunity threshold to block mumps virus transmission is estimated ≥86%. Seroprotection rates at 3 years after an MMR3 dose are therefore close to the level needed for herd immunity, and could be sufficient to limit widespread of the virus and reduce the risk of a mumps outbreak.

In 2016, the effectiveness of an MMR3 dose for outbreak control was investigated. The mumps attack rate was lower among students who had received 3 vaccine doses than among those who had received 2 doses (6.7 vs. 14.5 cases per 1000 population) [9]. In line with this, in 2017, a total of 26 cases of mumps occurred among 140 military recruits in a single compound. The attack rate was 86% among the soldiers without any MMR vaccination, 15% among those who had received two vaccine doses, while the attack rate was 8% among those who had received three vaccine doses. There was a trend that the attack rate was higher in soldiers who had received the third dose more than one year before the outbreak [28]. Apparently, individuals who received an MMR3 dose a few years ago may already be susceptible to mumps in a setting with high virus exposure risk.

An MMR vaccine booster dose, as intervention to control a mumps outbreak, should be given in the early phase of an outbreak to persons at risk for mumps, such as young adults that have been vaccinated for longer time ago. Apart from an additional MMR dose, the development of a new polyvalent mumps vaccine combining diverse genotypes including circulating outbreak strains have been suggested to provide a solution for better and long-term protection against mumps [29,30,31].

Interestingly, the lower the mumps-specific antibody levels were before vaccination, the stronger was the increase in antibodies after an MMR3 dose. Therefore, the immunity of persons with subprotective antibody levels was strongly boosted. This is line with previous findings from another group that found that individuals with the lowest baseline virus-neutralization antibody titers had the largest increase in seropositivity of antibodies measured by ELISA after a third MMR vaccine dose [32]. Apart from mumps, also immunity against measles and rubella is boosted by an additional MMR dose. Prior to an MMR3 dose, seroprotection rates for measles and rubella were 97% and 95%, respectively. Three years after an MMR3 dose, 100% of the participants were still seroprotected to measles and rubella according to the internationally agreed antibody cutoff levels for clinical protection. Especially health care professionals with declining mumps immunity who treat measles cases may benefit from an extra MMR3 dose [33,34]. Additionally, an MMR3 dose may be beneficial for nonpregnant postpubertal women with waning vaccine-induced measles antibody levels [33,34]. Children born from mothers in which antibody levels were significantly boosted will benefit from a longer duration of protection by maternal antibodies. In line with this, nonpregnant postpubertal women with low rubella titers may benefit from an MMR3 dose as well, by the induction of maternal antibodies providing protection to their future infants against congenital rubella syndrome. Nevertheless, at this moment there is no urgent reason to implement an extra MMR vaccine dose on a large scale for these groups to protect against measles or rubella.

In conclusion, three years after an MMR3 dose, mumps-specific antibody levels were still slightly higher compared to pre-vaccination levels. Moreover, all subjects had antibody levels to measles and rubella above the internationally agreed antibody cutoff levels for clinical protection. Our data support the hypothesis that an additional MMR dose boosts the immunity to the mumps virus. Therefore, we recommend to consider an MMR3 dose offered early during a mumps outbreak. An extra MMR vaccination is expected to reduce the number of susceptible individuals and thereby reduce the risk of onward virus transmission during an ongoing mumps outbreak. Furthermore, an MMR3 dose in young adults may have additional beneficial effects for protection against measles and rubella virus infection due to increased antibody levels lasting longer than might have been expected.

## Data Availability

The raw data supporting the conclusions of this article will be made available by the authors upon request, with consideration of the participants’ privacy and ethical rights. The setup and application of the linear mixed model were performed using R, version 3.4.3 [22] and visualization was performed using package ggplot2. This is an open-source data visualization package for the statistical programming language R [23].

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
