# Peer review of "Antibody Levels at 3-Years Follow-Up of a Third Dose of Measles-Mumps-Rubella Vaccine in Young Adults"

_vaccines, 2022, doi:10.3390/vaccines10010132_

Round 1
Reviewer 1 Report
This manuscript was written about the clinical importance of MMR3 vaccination. It is well written and the result was understandable. Several points are better to be modified.
- Are there any disadvantages to increasing the number of vaccine doses, which should be mentioned in the discussion?
- The text in figure is hard to understand. It should be corrected for clarity.
Author Response
Reviewer #1:
Comments and Suggestions for Authors
This manuscript was written about the clinical importance of MMR3 vaccination. It is well written and the result was understandable. Several points are better to be modified.
Author’s reply on comments of Reviewer #1:
We thank the reviewer for his/her compliments on the clinical importance of our study and that the manuscript is well-written with understandable results.
Are there any disadvantages to increasing the number of vaccine doses, which should be mentioned in the discussion?
A major disadvantage of an extra vaccine doses are the costs and the possible limited willingness of receiving an extra vaccine dose, as mumps is generally perceived as mild disease. It is most likely not cost effective and also not necessary to target the entire age group for an extra vaccine dose. Therefore, we recommend to consider an MMR3 dose offered early during a mumps outbreak as it is to expect to reduce the number of susceptible individuals and thereby reduce the risk of onward virus transmission. We have now more explicitly mentioned this at the end of the discussion section.
The text in figure is hard to understand. It should be corrected for clarity.
We have now enlarged the figures, and increased the resolution. In addition, we have used a larger character size for the figures 1 and 2. Hopefully, the text in the figures is now easier to read.
Reviewer 2 Report
A nice clear study that conveys an important and compelling message.
With much attention being placed upon the benefits of single of even double COVID booster vaccines some attention has been diverted away from the great benefits associated with the long-used MMR vaccines that provide such wonderful protection against measles, mumps and rubella. This fine study sought to provide data on the benefits of employing a third MMR (MMR3) dose particularly for providing additional protection to those individuals that have become susceptible to mumps virus infection during outbreaks.
The paper provides a succinct, but adequate, introduction to justify the need for this investigation. Sufficient relevant literature is cited, some carried out by the authors, to justify need for the investigation.
The materials and methods describe the immunization background for the participants that all had received MMR vaccines at age of 14 months and 9 years as part of a National Immunization Program operating in the Netherlands since 1987.
I suggest adding, at around L73, the number of initial participants, n=147, that first appears later in the results at L152. Admittedly, later in the Results at L153, it is revealed that the enduring participants remaining in the study after the MMR3 vaccine was 81% of the 147 young adults and the details of the gender are included at L156.
Having re-read these sections several times I suggest adding a time line or flow diagram to summarise the treatments and number of participants at each stage of the study which would help convey the design quite well.
The statistical treatment of data appear to be quite well chosen and effectively used throughout the study with additional explanations being provided for the statistical handling of data.
Results are presented in a methodical and effective manner with clear Figures and legends. Fig 1 and legend is effective in explaining and depicting the dynamic course in IgG and virus neutralizing antibodies for mumps virus after participants received MMR3. This is followed by and examination of the dynamic course of IgG antibody responses to measles, mumps and rubella virus after participants received MMR3.
In scatterplot form Fig 3 depicts IgG and virus neutralizing antibody levels to mumps and concentrations of IgG antibodies against rubella and measles both before and after administration of MMR3. This legend, in my opinion, is necessarily quite long but fully describes the data presented.
The discussion is relatively brief however the essential findings of the study are effectively stated and considered in light of several important references that are also cited. Upon viewing the data presented for this study the reader should agree with the conclusion reached by the authors that and MMR3 dose should be offered early during any mumps outbreak and that an MMR3 dose being recommended and made available to young adults would be of great benefit enabling enhanced protection against measles and rubella virus infection. The latter benefit would see infants of postpubertal women, in particular those with low rubella antibody titres, gaining enhanced protection against the preventable congenital rubella syndrome.
A nice clear study that provides compelling evidence that the administration of MMR3 vaccine can have important medical benefits, in particular for use during mumps outbreaks and for enhanced protection of women with low rubella antibody titres who are planning pregnancy.
Suggestions
L23 Abstract: Maybe use something a little stronger and more definite to support the recommendation - that the MMR3 should be given more routinely to those that have become susceptible to mumps virus infections during outbreaks. Eg rather than using "..support the assumption..." you could use "..support the recommendation that ...
or "...support the view that....”
L120 Better to use volume rather than amount
Author Response
Reviewer #2:
Comments and Suggestions for Authors
A nice clear study that conveys an important and compelling message.
Author’s reply on comments of Reviewer #2:
We thank the reviewer for his/her compliments on our study and compelling message of the manuscript.
With much attention being placed upon the benefits of single of even double COVID booster vaccines some attention has been diverted away from the great benefits associated with the long-used MMR vaccines that provide such wonderful protection against measles, mumps and rubella. This fine study sought to provide data on the benefits of employing a third MMR (MMR3) dose particularly for providing additional protection to those individuals that have become susceptible to mumps virus infection during outbreaks.
The paper provides a succinct, but adequate, introduction to justify the need for this investigation. Sufficient relevant literature is cited, some carried out by the authors, to justify need for the investigation.
The materials and methods describe the immunization background for the participants that all had received MMR vaccines at age of 14 months and 9 years as part of a National Immunization Program operating in the Netherlands since 1987.
I suggest adding, at around L73, the number of initial participants, n=147, that first appears later in the results at L152. Admittedly, later in the Results at L153, it is revealed that the enduring participants remaining in the study after the MMR3 vaccine was 81% of the 147 young adults and the details of the gender are included at L156. Having re-read these sections several times I suggest adding a time line or flow diagram to summarise the treatments and number of participants at each stage of the study which would help convey the design quite well.
As suggested by the reviewer, we have added the initial number of participants in the Methods section. We also provided a flow scheme for the treatments/sampling and number of participants at each stage of the study (Supplementary Figure 1).
The statistical treatment of data appear to be quite well chosen and effectively used throughout the study with additional explanations being provided for the statistical handling of data.
Results are presented in a methodical and effective manner with clear Figures and legends. Fig 1 and legend is effective in explaining and depicting the dynamic course in IgG and virus neutralizing antibodies for mumps virus after participants received MMR3. This is followed by and examination of the dynamic course of IgG antibody responses to measles, mumps and rubella virus after participants received MMR3.
In scatterplot form Fig 3 depicts IgG and virus neutralizing antibody levels to mumps and concentrations of IgG antibodies against rubella and measles both before and after administration of MMR3. This legend, in my opinion, is necessarily quite long but fully describes the data presented.
The discussion is relatively brief however the essential findings of the study are effectively stated and considered in light of several important references that are also cited. Upon viewing the data presented for this study the reader should agree with the conclusion reached by the authors that and MMR3 dose should be offered early during any mumps outbreak and that an MMR3 dose being recommended and made available to young adults would be of great benefit enabling enhanced protection against measles and rubella virus infection. The latter benefit would see infants of postpubertal women, in particular those with low rubella antibody titres, gaining enhanced protection against the preventable congenital rubella syndrome. A nice clear study that provides compelling evidence that the administration of MMR3 vaccine can have important medical benefits, in particular for use during mumps outbreaks and for enhanced protection of women with low rubella antibody titres who are planning pregnancy.
Suggestions: L23 Abstract: Maybe use something a little stronger and more definite to support the recommendation - that the MMR3 should be given more routinely to those that have become susceptible to mumps virus infections during outbreaks. Eg rather than using "..support the assumption..." you could use "..support the recommendation that ... or "...support the view that....”
L120 Better to use volume rather than amount
We thank the reviewer for these suggestions which we have applied in the text accordingly.